# Safety and Satisfaction Analysis of Intravenous and Inhalational Conscious Sedation in a Geriatric Population Undergoing Oral Surgery

**DOI:** 10.3390/healthcare13020116

**Published:** 2025-01-09

**Authors:** Elizabeth Iglesias-Rodeiro, Pedro Luis Ruiz-Sáenz, Cristina Madrigal Martínez-Pereda, Cristina Barona-Dorado, Fernando Fernández-Cáliz, Natalia Martínez-Rodríguez

**Affiliations:** 1Department of Dental Clinical Specialties, Faculty of Dentistry, Complutense University of Madrid, 28040 Madrid, Spain; elizigle@ucm.es (E.I.-R.); cmadriga@ucm.es (C.M.M.-P.); fernanfe@ucm.es (F.F.-C.); nataliamartinez@ucm.es (N.M.-R.); 2Department of Odontology, Central Hospital of the Red Cross of Madrid, 28003 Madrid, Spain; drruizsaenz@gmail.com; 3Surgical and Implant Therapies in the Oral Cavity Research Group, Complutense University, 28040 Madrid, Spain

**Keywords:** conscious sedation dentistry, midazolam, sevoflurane, hemodynamic changes, complications, gerodontology

## Abstract

**Aim**: The objective of this research was to compare two methods of conscious sedation (midazolam vs. sevoflurane) used for performing oral surgical procedures in the older adult population by analyzing dental treatment-related anxiety levels, the quality of sedation, and potential hemodynamic changes during the interventions, as well as post-recovery symptoms and patient satisfaction levels. **Methods**: A total of 104 patients underwent oral surgery with intravenous (age: 84.00 ± 8.410; 17 men and 36 women) and inhalation conscious sedation (age: 82.73 ± 8.809; 18 men and 33 women). Anxiety levels were recorded using the Modified Corah Dental Anxiety Scale (MDAS) and the Hamilton Anxiety Rating Scale (HARS). During the intervention, the heart rate, systolic/diastolic blood pressure, oxygen saturation, episodes of hypoxia (<90%SpO2), sedation duration, and Sedation–Agitation Scale (SAS) values were monitored. Upon completion of their treatment, all patients completed three questionnaires which evaluated their recall of the intervention, postoperative symptoms, and recommendation of the sedation method used. **Results**: Anxiety levels were slightly higher in the midazolam group (MDAS score: 9.60 ± 5.849; HARS score: 27.19 ± 6.802) compared to the sevoflurane group (MDAS score: 8.37 ± 5.063; HARS score: 24.92 ± 5.199), with no statistical significance (MDAS, *p* = 0.453/HARS, *p* = 0.065). No significant differences in the analyzed hemodynamic parameters were observed between the two groups. However, SAS values were significantly higher in the sevoflurane group (*p* = 0.006), while the sedation duration was longer in the midazolam group (*p* = 0.000). Postoperative symptoms, particularly gastrointestinal disturbances and dry mouth, were significantly more prevalent in the midazolam group, while there were no differences in neurological symptoms between the two groups. The majority of patients expressed a preference for either of these sedation methods. **Conclusions**: Sedation with agents such as midazolam and sevoflurane appears to be quite safe, given the absence of relevant hemodynamic changes. Midazolam has been shown to be effective for a longer duration, as well as to have a lower risk of developing greater digestive problems during the postoperative period. On the contrary, sevoflurane produced a greater sensation of dry mouth. Both sedative agents are perceived by the older adult population as satisfactory, recommending their use.

## 1. Introduction

An aging population is an unquestionable reality, with the proportion of older adults in the population expected to reach 22% by 2050, as Hazara [1] points out. This increase will be accompanied by a greater prevalence of systemic diseases, and increased pharmacological treatments will make the older adult population more vulnerable to oral diseases such as periodontal disease, dental caries, and even precancerous lesions [2,3].

Coll et al. [4] report that the prevalence of root cavities was more than double in older adults (36%) compared to younger adults according to data collected in 2020 by the National Center for Health Statistics of the United States. Likewise, the prevalence of periodontitis also increased with age, with 64% of older adults experiencing moderate or severe periodontitis, compared to less than 38% of younger patients.

Although aging has no or very limited effects on the treatment of dental cavities and periodontal diseases, it is important to note that there are different factors involved, such as the use of medications, reduced salivary secretion, deteriorated fixed and removable dental prostheses, and a decrease in one’s ability to maintain an adequate oral hygiene, which can have direct consequences on oral health and consequently lead to tooth loss. Therefore, the treatment of cavities and periodontal diseases is one of the most common treatments in this population group [5,6].

Tooth extractions may differ in terms of technique and the occurrence of complications, and differences between population groups must also be taken into consideration. There are greater risks for older adults, due to their comorbidities, in cases of pathologies such as diabetes or in situations of impacted teeth [7,8].

Oral surgery procedures are common in dental practices and rarely pose a significant risk to the patient. However, they are often perceived as painful and invasive, leading to many patients experiencing dental anxiety when facing such procedures [9]. This anxiety frequently results in the postponement or avoidance of necessary treatments, which can lead to a worsening of oral health. This situation can have a particularly significant impact on quality of life in the geriatric population [10].

Anxiety is defined as “the apprehensive anticipation of future harm” [11], with varying levels ranging from “none” to “extreme”, depending on the diagnostic scale implemented [12]. Dental anxiety or fear has a global prevalence of 15.3%, affecting women more than men, in a 2:1 ratio [13]. The development of more advanced techniques ensuring simpler, less painful, and less invasive procedures has contributed to a reduction in the incidence of dental anxiety [14]. However, older adults represent an exception, as they often present with systemic comorbidities, in addition to oral health problems. Not only do these comorbidities influence the dental treatment itself [15], but also the anxiety generated by the treatment can have a physiological impact. This impact includes the release of endogenous catecholamines and cortisol, which are associated with dental anxiety and stress, leading to increases in blood pressure and heart rate—factors that may result in cardiovascular decompensation in these patients [16,17].

Dentists play a crucial role in identifying patients who may suffer from anxiety. In an effort to minimize this anxiety, complementary procedures are sought to facilitate the patients’ willingness to undergo dental treatments, thereby improving their oral health and, consequently, their quality of life [18].

Sedation has emerged as an option to address this issue, as it not only allows for a reduction in patient anxiety and fear but also improves their perception of the treatment, enhancing cooperation and making the clinician’s work easier [19]. Sedation administered by an anesthesiologist is a common procedure in hospital-based oral surgery departments. However, it also presents an interesting option as an alternative to general anesthesia in dental offices, which requires more resources and longer recovery times. Additionally, sedation provides the advantage of not necessitating airway management, as spontaneous ventilation and cardiovascular function are maintained [20,21].

Various drugs can be used for sedation during oral surgical procedures. One of the most commonly used drugs for surgical interventions is midazolam, a fast-acting benzodiazepine with a short elimination time, which is particularly favored for its sedative, anxiolytic, and amnesic effects [22]. Its effects can be antagonized with flumazenil, if necessary [23]. However, its pharmacological effects often extend beyond the procedure, and for patient safety, postoperative monitoring is recommended [24,25,26].

Another sedative agent used is sevoflurane, administered via inhalation. It has rapid induction and recovery times due to its low blood solubility, with minimal effects on respiratory and cardiovascular function [27]. Additionally, its non-invasive application (without injections) is advantageous for anxious patients, particularly those with multiple pathologies, as well as for geriatric and pediatric patients, in whom it is most frequently used [28,29].

Although these sedative agents can be administered in combination with each other or alongside other medications, their isolated use is often preferred to limit the risk of adverse events, particularly in polymedicated geriatric patients, where the likelihood of drug interactions may be increased [30,31,32].

In the dental field, interest in conscious sedation is increasing across different population groups, particularly in the geriatric population, due to the aging of the general population. In this regard, the scientific literature specifically comparing the use of inhalation sedation with sevoflurane and intravenous sedation with midazolam in geriatric populations is limited. Thus, this study is justified. Our null hypothesis states that neither sedation method will present differences in terms of complications and the patients’ degree of satisfaction. Therefore, the objective of this research was to compare these two conscious sedation methods commonly used in oral surgical procedures in the older adult population. This study first examined patients’ levels of anxiety regarding oral surgical procedures, the quality of the sedation, and potential hemodynamic changes during the interventions. Next, it evaluated the symptoms during recovery, as well as the perceived level of patient satisfaction.

## 2. Materials and Methods

### 2.1. Study Design

A prospective clinical study of two cohorts was conducted, following the model proposed by the CONSORT statement for clinical trials. All participating patients signed an informed consent form. This study was approved by the Ethics Committee of La Paz University Clinical Hospital in Madrid (HULP. PI-5005, Ref: 07/111114.9/22). The inclusion and exclusion criteria are presented in Table 1.

### 2.2. Participants

A total of 189 patients from geriatric centers in the Community of Madrid were referred to the Dental Service of the Central Hospital of the Red Cross in Madrid by their dentists to undergo multiple dental extractions. These patients were recruited by the person responsible for the service (P.L.R.-S.) from February 2022 to June 2023. To calculate the necessary sample size for an independent samples *t*-test, the SPSS 30 program was used, based on the hypothesis of observing a difference of 0.14 between the groups with a deviation of 0.25, resulting in a sample of 52 patients per group (with a power of 80% and a significance of 0.05).

The patients were informed about the objectives of this study and the sedation anesthetic procedure, with 106 patients agreeing to participate. The patients were randomized by one of the investigators (E.I.-R.) using random-number-generating statistical software and opaque envelopes, with another investigator (P.L.R.-S.) assigning each patient to one of two groups: group 1 (intravenous sedation with midazolam) and group 2 (inhalation sedation with sevoflurane).

### 2.3. Preoperative Records

Two days prior to the surgical intervention, the patients were evaluated by a single anesthesiologist responsible for sedation, who recorded each patient’s age, gender, height (in centimeters), and weight (in kilograms) and classified them as ASA III or IV. The anesthetic and surgical procedures were explained to the patients, and any potential questions were addressed.

Subsequently, an independent investigator (E.I.-R.) administered two preoperative anxiety questionnaires.

The Modified Dental Anxiety Scale (MDAS) [33] is a 5-item scale in which each item is scored as follows: 0: relaxed, not anxious; 1: slightly anxious; 2: fairly anxious; 3: very anxious and uneasy; and 4: extremely anxious. The total score is interpreted as follows: a score of 9 points indicates no or mild anxiety; 9–12 points indicate moderate anxiety; 13–14 points indicate high anxiety; and >15 points indicate severe anxiety levels (Table 2).

The Hamilton Anxiety Rating Scale (HARS) consists of 14 items (Table 3), each scored as follows: 0: absent; 1: mild; 2: moderate; 3: severe; and 4: very severe or incapacitating. The total possible score ranges from 0 to 56 points, with a score of >14 considered indicative of anxiety that may require treatment [34,35].

### 2.4. Anesthetic Procedure and Intraoperative Records

Patients in the midazolam group received an intravenous dose of 0.05 mg/kg. Patients in the sevoflurane group were administered an 8% concentration via a full facial mask until loss of consciousness, after which the mask was replaced with a nasal cannula. For the remainder of the procedure, oxygen was maintained at a flow rate of 2 L/min at 100%, with sevoflurane at a concentration of 1–1.5%.

Once the desired sedation level was achieved, the sedation start time was recorded, and local anesthesia (articaine (4%) with 1:100,000 adrenaline) was administered. Five minutes before the surgical procedure was completed, the surgeon notified the anesthesiologist to begin reversing the sedation. During the intervention, patients were continuously monitored, with records of the heart rate, systolic/diastolic pressure, oxygen saturation, episodes of hypoxia (<90%SpO2), sedation time, and the Sedation–Agitation Scale (SAS) values [36] used to assess sedation quality (Table 4).

Upon completion of the intervention, patients were transferred to the post-anesthesia recovery unit and, once conscious with stable cardiorespiratory values, were discharged by the anesthesiologist.

### 2.5. Postoperative Records

All patients were given three questionnaires to complete and return seven days later, coinciding with their postoperative follow-up.

The first questionnaire assessed the patients’ recall of the procedure using a 4-point Likert scale, with the following scoring: 1: no recall; 2: a little; 3: most of it, and 4: all of it.

The second questionnaire recorded the symptoms during recovery, which had been previously explained to the patients, and the presence of neurological and gastrointestinal symptoms, dry mouth, and postoperative bleeding.

The third and final questionnaire assessed the recommendation of sedation treatment, for which the answer was yes or no.

In order to eliminate bias, the results of these questionnaires were collected by two different researchers (C.M.M.-P. and C.B.-D).

### 2.6. Statistical Analysis

For statistical analysis, IBM SPSS Statistics for Windows (version 27.0, IBM Corp., Armonk, NY, USA) was used to provide a detailed description of the data in frequencies and percentages.

The normality of the sample was assessed using the Kolmogorov–Smirnov and Shapiro–Wilk tests. Descriptive statistics for all variables were calculated for the total sample and for each group. Qualitative variables were compared using the chi-square test or Fisher’s exact test, and for quantitative variables, the Mann–Whitney test was performed, as normality was rejected in various variables. Statistical significance was set at *p* ≤ 0.05.

## 3. Results

Of the 106 participating patients, two from group 2 (sevoflurane) did not complete the postoperative questionnaires and were excluded from the study, leaving the two groups as follows: group 1 (midazolam) with 53 patients and group 2 (sevoflurane) with 51 patients (Figure 1).

### 3.1. Sample Characteristics

Table 5 presents the basic clinical and demographic characteristics of the patients in both groups. Age, weight, height, and gender were similar between the two groups, without significant differences.

### 3.2. Preoperative Records (Anxiety Scales)

The mean scores of patient responses to the MDAS and HARS questionnaires were slightly higher in the midazolam group. However, these differences were not statistically significant (Table 6).

### 3.3. Intraoperative Records

Table 7 shows that, during treatment, all hemodynamic parameters remained within normal limits, with no differences between the two groups. No episodes of hypoxia were recorded. The Sedation–Agitation Scale (SAS) values also remained within normal parameters, with slightly higher values in the sevoflurane group. Surgical time was similar in both groups, while sedation time was significantly longer in the midazolam group (*p* < 0.001).

### 3.4. Postoperative Records

Table 8 shows the values obtained from the three questionnaires completed by the patients. Regarding their recall of the procedure, significant differences were found between the two groups, with patients in the sevoflurane group remembering the procedure more clearly. In the questionnaire on symptoms during recovery, nearly half of the midazolam group reported experiencing gastrointestinal symptoms (nausea, vomiting, abdominal pain, and gas/flatulence), while no symptoms were reported in the sevoflurane group (*p* < 0.001).

Central nervous system symptoms (fatigue/sleepiness, headache, dizziness, and confusion) were similar in both groups, disappearing within 36 h post-treatment. Seven patients in the sevoflurane group reported transient bleeding, compared to none in the midazolam group (*p* = 0.005). The sensation of dry mouth was more prevalent in the midazolam group (32 patients) compared to the sevoflurane group (15 patients) (*p* = 0.002).

In the third questionnaire, concerning the recommendation of sedation treatment, 41 patients from the midazolam group and 32 from the sevoflurane group responded affirmatively, with no significant statistical difference between the groups.

## 4. Discussion

Dental anxiety represents a barrier for many patients undergoing dental procedures, and it can be identified using various scales, including the Modified Dental Anxiety Scale (MDAS) and the Hamilton Anxiety Rating Scale (HARS).

The results of this study showed that the groups were homogeneous regarding their demographic characteristics. Additionally, anxiety levels in the older adult population were generally not excessively high, aligning with the findings of Tarrosh et al. [37], who evaluated the reliability of the MDAS and its correlation with demographic factors such as age and gender. The authors found that older adult patients exhibited lower levels of anxiety compared to younger age groups, while women showed higher anxiety levels than men; these results were consistent with those of Dadalti et al. [38].

The anxiety levels assessed using the HARS were slightly higher in the midazolam group, though these differences were not statistically significant, results that are similar to those of other studies comparing different sedation methods [39,40].

Another objective was to record potential hemodynamic changes during sedation. The American Dental Association (ADA) establishes sedation-monitoring parameters in dentistry, including the sedation level, oxygenation, ventilation, circulation, and documentation of all drugs administered, including local anesthetics and the duration of their administration [41].

In this study, the heart rate, systolic/diastolic blood pressure, and oxygen saturation values obtained with sevoflurane were similar to those obtained with midazolam. These results are consistent with the findings of Nishiaki et al. [42] and Morimoto et al. [43], who also found no significant differences between these parameters.

Regarding the quality of sedation assessed using the Sedation–Agitation Scale (SAS), statistically significant differences were observed between the two groups, with the sevoflurane group showing slightly higher values on the scale compared to the midazolam group. Similar results were obtained by McLaren et al. [44], whose patients sedated with midazolam showed lower SAS values, highlighting midazolam’s capacity to reduce anxiety during surgical procedures under sedation. Similarly, Kim et al. [45] showed that SAS values associated with sevoflurane use alone were higher after the completion of sedation compared to when combined with alpha-2 adrenergic receptor agonists, due to their sedative effect.

Regarding patient satisfaction, as measured using the recall and recommendation questionnaires, significant differences were found between the two groups. Patients sedated with midazolam showed greater overall satisfaction, likely due to its amnesic effect, as noted by Sugimura et al. [32] and Chi [46]. Wang et al. [47] reported that midazolam requires less time to take effect, leading to higher satisfaction scores, aligning with the findings of Gargallo-Albiol et al. [48], wherein patients undergoing oral surgery under midazolam sedation reported greater satisfaction compared to those treated with local anesthesia. Conversely, Ohkushi et al. [49] found that patients reported greater satisfaction with propofol compared to sevoflurane, as they experience discomfort due to the taste and smell of the mask used to administer sevoflurane via inhalation.

In summary, patients are generally satisfied with sedation for anxiety control and reduction. This translates into greater effectiveness and satisfaction for professionals, as evidenced by Pourabbas et al. [50], who concluded that sedation is an effective method for reducing anxiety and increasing patient and professional satisfaction.

Regarding postoperative symptoms, significant differences were observed in gastrointestinal symptoms and dry mouth, which were more prevalent in the midazolam group compared to the sevoflurane group. Similarly, two meta-analyses, by Kim et al. [27] and Jerath et al. [28], comparing intravenous propofol sedation with sevoflurane sedation in patients in the ICU highlighted sevoflurane’s advantages in the faster recovery of consciousness and the lower incidence of nausea and vomiting during recovery.

Conversely, in a meta-analysis, Grant et al. [51] concluded that midazolam sedation in adult patients undergoing various surgical procedures is associated with a reduction in postoperative nausea and vomiting. Similarly, in a comparative study of intravenous versus inhalation sedation in adult patients undergoing major and minor oral and maxillofacial surgery, Şimşek et al. [52] found that patients sedated with sevoflurane had a higher incidence of nausea and vomiting than those treated with propofol. These findings may be due to short postoperative follow-up periods in inhalation sedation, indicating the need for further studies with longer follow-up periods.

Postoperative bleeding increased in the sevoflurane group, likely due to sevoflurane’s known inhibitory effect on platelet aggregation, as demonstrated by Doğan et al. [53] and later confirmed by Liang et al. [54] in an in vitro study on coagulation times. However, no differences were observed in neurological symptoms between the two groups.

The significant differences in sedation time, which was shorter in the sevoflurane group, can be easily explained by the faster recovery time associated with sevoflurane use [55], as demonstrated by Mei et al. [56]. The results showed that neither the faster recovery time nor the lower incidence of gastrointestinal symptoms or dry mouth appeared to positively influence higher satisfaction for patients treated with sevoflurane. No benefits were seen in patient recovery, as the occurrence of nervous system symptoms (fatigue/somnolence, headache, dizziness, and confusion) was similar in both groups.

In summary, the results of this research confirm its strength and clinical contribution, given the absence of dentistry research that compares both sedative agents within the older adult population. These findings constitute a first step; however, further research directions must be explored in order to evaluate whether these results are upheld. The first would be to assess whether the difference in the mode of administration of the sedative agent (intravenous vs. inhalation) could change the results if a study were carried out on sedative agents administered in the same way. For characteristic examples of this, see several studies on propofol and midazolam, which achieved high levels of satisfaction [57,58,59].

A limitation of this study is the type of surgical treatment performed. In dentistry and especially in older adult patients, edentulism is common, which is why rehabilitation with osseointegrated implants is a common therapeutic option. In the latter case, soft tissues and bone tissue are manipulated to a higher degree than they were in this study, so it would be desirable to be able to perform this type of rehabilitation for this indication, as authors such as Kawaaieta [60] and McCrea [61] did.

A final limitation is that these results may be similar within this population group if the patients had cognitive disorders [62].

To conclude, it can be generalized that sedation procedures can be very valuable tools in the field of oral surgery since they are safe for the patient, given the low incidence of adverse effects. The older adult population may be particularly complex to treat in many cases due to systemic and cognitive conditions, which is why sedation may be beneficial. It is essential that professionals applying conscious sedation methods are well versed in the guidelines for the use of different medications and are properly trained to administer them safely and to address potential adverse events.

## 5. Conclusions

Sedation with agents such as midazolam and sevoflurane appears to be quite safe, given the absence of relevant hemodynamic changes. Midazolam has been shown to be effective for a longer duration, as well as to cause more digestive problems during the postoperative period. On the contrary, sevoflurane produced a greater sensation of dry mouth. Both sedative agents have been perceived by the older adult population as satisfactory, recommending their use, so in this aspect, the null hypothesis raised at the beginning of this research would be confirmed.

## Figures and Tables

**Figure 1 healthcare-13-00116-f001:**
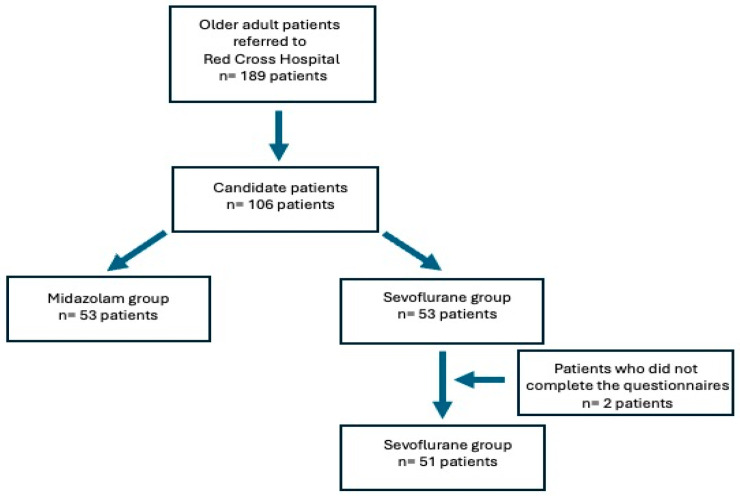
Patient selection flow chart.

**Table 1 healthcare-13-00116-t001:** Inclusion and exclusion criteria for patients admitted to the study.

Inclusion Criteria	Exclusion Criteria
Male and female patients over 60 years old.	Patients who did not wish to participate in this study.
2. ASA III and ASA IV patients (patients with severe systemic disease and patients with severe systemic disease that is a constant threat to life, respectively).	2. Patients who did not complete any of the questionnaires.
3. Patients requiring oral surgical treatment.	3. Patients with a history of hypersensitivity to sevoflurane or other halogenated anesthetics.
4. Patients undergoing sedation for 25 to 45 min. 5. Patients who completed pre- and post-treatment questionnaires.	4. Patients with hypersensitivity to benzodiazepines. 5. Patients for whom sevoflurane may be contraindicated, such as those with neuromuscular diseases or those on selective monoamine oxidase inhibitors (MAOIs). 6. Patients for whom midazolam may be contraindicated, such as those with myasthenia, severe hepatic or respiratory insufficiency, sleep apnea syndrome, hepatic or renal dysfunction, or narrow-angle glaucoma.

**Table 2 healthcare-13-00116-t002:** Modified Dental Anxiety Scale (MDAS) five-item questionnaire.

Corah Anxiety Questionnaire	
If you were going to the dentist tomorrow for a checkup, how would you feel about it?	Relaxed, not anxious Slightly anxious Fairly anxious Very anxious and uneasy Extremely anxious (sweating, tachycardia, feeling severely unwell)
2. While waiting for your turn in the dentist’s waiting room, how do you feel?	Relaxed, not anxious Slightly anxious Fairly anxious Very anxious and uneasy Extremely anxious (sweating, tachycardia, feeling severely unwell)
3. When sitting in the dentist’s chair while the dentist prepares the drill to start working on your teeth, how do you feel?	Relaxed, not anxious Slightly anxious Fairly anxious Very anxious and uneasy Extremely anxious (sweating, tachycardia, feeling severely unwell)
4. Imagine you are in the dentist’s chair for a cleaning. While waiting, the dentist or hygienist pulls out the instruments to scrape your teeth around the gums. How do you feel?	Relaxed, not anxious Slightly anxious Fairly anxious Very anxious and uneasy Extremely anxious (sweating, tachycardia, feeling severely unwell)
5. If you were to receive a local anesthetic injection for your dental treatment, how would you feel?	Relaxed, not anxious Slightly anxious Fairly anxious Very anxious and uneasy Extremely anxious (sweating, tachycardia, feeling severely unwell)

**Table 3 healthcare-13-00116-t003:** Hamilton Anxiety Rating Scale (HARS).

Symptoms of Anxiety States
1. Anxious mood Worries, anticipation of the worst, apprehension (fearful anticipation), irritability
2. Tension Feeling tense, fatigability, inability to relax, startled reactions, crying easily, tremors, restlessness
3. Fears Fear of the dark, strangers, being alone, large animals, traffic, and crowds
4. Insomnia Difficulty falling asleep, interrupted sleep, unsatisfactory sleep, fatigue upon waking
5. Intellectual (cognitive) symptoms Difficulty concentrating, poor memory
6. Depressed mood Loss of interest, dissatisfaction with hobbies, depression, early waking, mood changes during the day
7. Somatic (muscular) symptoms Muscle aches and pains, muscle stiffness, muscle twitches, clonic movements, teeth grinding, tremulous voice
8. Somatic (sensory) symptoms Tinnitus, blurred vision, hot flashes and chills, feelings of weakness, tingling sensation
9. Cardiovascular symptoms Tachycardia, palpitations, chest pain, vascular pulsations, fainting sensation, extrasystole
10. Respiratory symptoms Chest tightness, feeling of suffocation, sighing, dyspnea
11. Gastrointestinal symptoms Difficulty swallowing, gas, dyspepsia (pain before or after eating), burning sensation, feeling of fullness, watery vomiting, nausea, vomiting, empty stomach sensation, slow digestion, borborygmi (intestinal sounds), diarrhea, weight loss, constipation
12. Genitourinary symptoms Frequent urination, urgent urination, amenorrhea, menorrhagia, onset of frigidity, premature ejaculation, absence of erection, impotence
13. Autonomic symptoms Dry mouth, flushing, pallor, tendency to sweat, dizziness, tension headaches, piloerection (goosebumps)
14. Behavior during the interview (general) Tense, not relaxed, nervous agitation (hands, clenched fingers, tics, fidgeting), restlessness (pacing), trembling hands, furrowed brow, taut facial muscles, increased muscle tone, sighing, pallor of the face Behavior during the interview (physiological) Swallowing saliva, belching, rest tachycardia, respiratory rate above 20 beats/min, vigorous tendon jerks, tremors, dilated pupils, exophthalmos (abnormal projection of the eyeball), sweating, eyelid twitches

**Table 4 healthcare-13-00116-t004:** Sedation–Agitation Scale (SAS).

Score	Classification	Description
7	Dangerous agitation	Pulls at the endotracheal tube or catheters, attacks the staff, attempts to jump from the bed.
6	Very agitated	Is unable to calm down despite verbal reassurance, requires physical restraint, bites the endotracheal tube.
5	Agitated	Is anxious or moderately agitated, attempts to sit up, calms down with verbal instructions.
4	Calm and cooperative	Is calm and easily arousable, follows simple commands.
3	Sedated	Shows a tendency to sleep, responds to verbal stimuli but falls back asleep, follows simple commands.
2	Very sedated	Responds to physical stimuli, is unable to communicate or follow commands, exhibits spontaneous movements.
1	Unresponsive	Shows minimal to no response to pain, no communication, or ability to follow commands.

**Table 5 healthcare-13-00116-t005:** Characteristics of the patients included in the study.

	Midazolam Group (*n* = 53)	Sevoflurane Group (*n* = 51)	*p*-Value
Age (years)	84.00 ± 8.410	82.73 ± 8.809	0.545
Weight (kg)	67.489 ± 14.0531	66.470 ± 14.389	0.471
Height (cm)	157.62 ± 8.221	157.65 ± 8.688	0.879
Gender	17 men/36 women	18 men/33 women	0.836

**Table 6 healthcare-13-00116-t006:** Anxiety scale scores.

	Midazolam Group (*n* = 53)	Sevoflurane Group (*n* = 51)	*p*-Value
Modified Dental Anxiety Scale (MDAS)	9.60 ± 5.849	8.37 ± 5.063	0.453
Hamilton Anxiety Rating Scale (HARS)	27.19 ± 6.802	24.92 ± 5.199	0.065

**Table 7 healthcare-13-00116-t007:** Intraoperative records.

Intraoperative Records	Midazolam Group (*n* = 53)	Sevoflurane Group (*n* = 51)	*p*-Value
Heart rate (ppm)	79.49 ± 15.092	76.06 ± 13.381	0.256
Systolic blood pressure (mmHg)	126.13 ± 23.824	126.14 ± 20.988	0.964
Diastolic blood pressure (mmHg)	71.04 ± 11.012	71.33 ± 10.318	0.951
Oxygen saturation	94.75 ± 2.377%	95.33 ± 2.304%	0.200
Sedation–Agitation Scale (SAS) score	3.36 ± 0.857	3.82 ± 0.684	0.006
Surgical time (min)	26.89± 5.235	26.30 ± 6.298	0.888
Sedation time (min)	40.26 ± 5.506	31.33 ± 6.154	0.001

**Table 8 healthcare-13-00116-t008:** Postoperative records.

Postoperative Records	Midazolam Group (*n* = 53)				Sevoflurane Group (*n* = 51)				*p*-Value
Procedure Recall	0:66%	1:34%	2:0%	3:0%	0:11.8%	1:43.1%	2:35.3%	3:9.8%	0.000
Postoperative symptoms	Digestive: 45.3%				Digestive: 0%				0.001
	Neurological: 94.3%				Neurological: 98%				0.618
	Hemorrhage: 0%				Hemorrhage: 13.7%				0.005
	Dry mouth: 60.4%				Dry mouth: 29.4%				0.002
Recommend the procedure	No: 22.64%				No: 37.25%				0.134
	Yes: 77.35%				Yes: 62.74%				

## Data Availability

The original contributions presented in this study are included in the article. Further inquiries can be directed to the corresponding authors.

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
