# Peer review of "Safety and Satisfaction Analysis of Intravenous and Inhalational Conscious Sedation in a Geriatric Population Undergoing Oral Surgery"

_healthcare, 2025, doi:10.3390/healthcare13020116_

Round 1

Reviewer 1 Report

Comments and Suggestions for Authors

I would like to congratulate the authors for their prospective clinical study on such a current and pertinent subject in geriatric population.

I think the article is well written with no factual errors with extensive and recent bibliography.

I will make some comments and suggestions in order to improve even more the quality of the manuscript:

-In introduction, the authors made a good review of literature and underlined the aim of the work, but I suggest they also mention the justification for the study, does it address a specific gap in the field?

-Concerning materials and methods: the methodology is very clear and well described in subtopics.

 I suggest the authors mention how did they calculate the sample size.

Any justification for the use of these 2 specific drugs, knowing that there are other pharmacological options?

- Results are clear and very well presented through the tables.

-Discussion is well done according to the results and supported by appropriate references. I really would like if you could add limitations of the study.

-Conclusions should respond to the objective proposed. “Therefore, the objective of this research was to compare these two conscious sedation methods employed for performing oral surgical procedures in an elderly population.”

Author Response

I would like to congratulate the authors for their prospective clinical study on such a current and pertinent subject in geriatric population.

I think the article is well written with no factual errors with extensive and recent bibliography. 

I will make some comments and suggestions in order to improve even more the quality of the manuscript:

-In introduction, the authors made a good review of literature and underlined the aim of the work, but I suggest they also mention the justification for the study, does it address a specific gap in the field?

In the dental field, interest in conscious sedation is increasing in different population groups, particularly in the geriatric population due to the aging of the general population. In this sense, the scientific literature that specifically compares the use of inhalation sedation with sevoflurane and intravenous sedation with midazolam in geriatric populations is limited, which is why the present study is justified.

-Concerning materials and methods: the methodology is very clear and well described in subtopics.

 I suggest the authors mention how did they calculate the sample size. 

To calculate the sample size, the SPSS 30 program was used for an independent samples T-test, based on the hypothesis of seeing a difference of 0.14 between the groups with a deviation of 0.25, resulting in a sample of 52 patients per group (power of 80% and significance of 0.05).

Any justification for the use of these 2 specific drugs, knowing that there are other pharmacological options?

For sedation, several medications may be used during oral surgical procedures. One of the most used drugs for surgical interventions is midazolam, a fast-acting benzodiazepine with a short elimination time, particularly favored for its sedative, anxiolytic and amnestic effects; and its effects can be antagonized by flumazenil. However, its pharmacological effects can extend beyond the procedure, so postoperative monitoring is recommended. Another sedative agent used is sevoflurane, administered by inhalation. It has rapid induction and recovery times due to its low blood solubility, with minimal effects on respiratory and cardiovascular function. Furthermore, its non-invasive application (without injections) is advantageous for anxious patients, particularly those with multiple pathologies, as well as for geriatric and pediatric patients, in whom it is used more frequently. Although these sedative agents can be administered in combination with each other or with other medications, their use alone is often preferred to limit the risk of adverse events, particularly in polypharmacy geriatric patients, where the likelihood of drug interactions may be increased.

- Results are clear and very well presented through the tables.

-Discussion is well done according to the results and supported by appropriate references. I really would like if you could add limitations of the study.

The following limitations have been included:

1.- The first of them would be to assess whether the difference in the route of administration of the sedative agent (intravenous vs inhalation) could change the results if the study were carried out between sedative agents administered in the same way. A typical example is some studies between propofol and midazolam that reach high levels of satisfaction.

2.- Another limitation that can be deduced is the type of surgical treatment performed in the present study. In Dentistry and in older adult patients, it is easy to find states of edentulism, which is why rehabilitation with osseointegrated implants is a frequent therapeutic option. In these cases, the manipulation of soft tissues and bone tissue is greater than that carried out in this research, so it would be desirable to be able to perform it in this indication, as authors such as Kawaai et al  and McCrea refer.

3.- A final limitation would be to consider whether these results could be similar, within this population group, if they were patients with cognitive disorders.

-Conclusions should respond to the objective proposed. “Therefore, the objective of this research was to compare these two conscious sedation methods employed for performing oral surgical procedures in an elderly population.”

After analyzing the results, it is concluded that sedation procedures can be very valuable tools in the field of oral surgery, since they offer safety for the patient, given the low incidence of adverse effects. The older adult population may be particularly complex in many cases due to systemic and cognitive conditions, which could benefit from sedation treatments. It is essential that professionals applying conscious sedation methods are well versed in the guidelines for the use of different medications and are properly trained to administer them safely and address potential adverse events.

Reviewer 2 Report

Comments and Suggestions for Authors

The paper is an interesting clinical trial on the comparison between midazolam and sevoflurane in conscious sedation for oral surgery procedures in elderly patients. The paper is well structured and presented, however there is little clarity regarding the power sample according to the assessed outcomes. Further it might be really interesting an evaluation between drugs within the same delivery as IV vs IV or inhalation vs inhalation. Overall the study seems eligible for the publication with minor changes.

Author Response

The paper is an interesting clinical trial on the comparison between midazolam and sevoflurane in conscious sedation for oral surgery procedures in elderly patients. The paper is well structured and presented, however there is little clarity regarding the power sample according to the assessed outcomes. Further it might be really interesting an evaluation between drugs within the same delivery as IV vs IV or inhalation vs inhalation. Overall the study seems eligible for the publication with minor changes.

To calculate the sample size, the SPSS 30 program was used for an independent samples T-test, based on the hypothesis of seeing a difference of 0.14 between the groups with a deviation of 0.25, resulting in a sample of 52 patients per group (power of 80% and significance of 0.05).

The following limitations have been included:

1.- The first of them would be to assess whether the difference in the route of administration of the sedative agent (intravenous vs inhalation) could change the results if the study were carried out between sedative agents administered in the same way. A typical example is some studies between propofol and midazolam that reach high levels of satisfaction.

Reviewer 3 Report

Comments and Suggestions for Authors

Dear Authors, Thank you for the opportunity to review your manuscript, "Intravenous vs Inhalation Conscious Sedation in Oral Surgery: A Safety and Satisfaction Analysis in a Geriatric Population." It is interesting and well-written. However, some corrections are necessary to improve its quality.

Title: You should report the article type in the title. Remove the full stop.

Introduction: Report what makes the study innovative, unique, and important. Why is it a contribution to the scientific literature?

Materials and Methods: Was the study registered? Please provide the registration number.

You should report what ASA III and IV mean. Introduce each abbreviation the first time you use it in the text.

Please report how bias was removed from the research.
Who generated the random allocation sequence and who enrolled the patients?

How did you determine the sample size?

Please report the articaine concentration.

Since you have followed the Consort Statement, you should provide the checklist as a supplementary file and add the Consort flow diagram in the text. The abstract should also be formatted according to these guidelines.

What are the research limitations? Please report.

Conclusion: You should summarize the main findings in this section.

Author Response

Dear Authors, Thank you for the opportunity to review your manuscript, "Intravenous vs Inhalation Conscious Sedation in Oral Surgery: A Safety and Satisfaction Analysis in a Geriatric Population." It is interesting and well-written. However, some corrections are necessary to improve its quality.

Title: You should report the article type in the title. Remove the full stop.

Safety and satisfaction analysis of intravenous and inhalational conscious sedations in a geriatric population undergoing oral surgery

Introduction: Report what makes the study innovative, unique, and important. Why is it a contribution to the scientific literature?

The reason this study is not only interesting but important is due to the fact that the scientific literature specifically comparing the use of inhalation sedation with sevoflurane and intravenous sedation with midazolam in geriatric populations is limited. Furthermore, in the dental field, interest in conscious sedation is increasing in different population groups, particularly in the geriatric population due to the aging of the general population.

Materials and Methods: Was the study registered? Please provide the registration number.

Ref: 07/111114.9/22.

You should report what ASA III and IV mean. Introduce each abbreviation the first time you use it in the text.

  1. We have included the description of ASA III and ASA IV patients:
  2. -ASA III: patients with severe systemic diseases

     -ASA IV: patients with severe systemic diseases that pose a constant threat to life.

Please report how bias was removed from the research. Who generated the random allocation sequence and who enrolled the patients?

With the intention of eliminating biases, we proceeded as follows:

The distribution of patients within the groups was carried out by randomization. Randomization was performed by one of the investigators (E.I.-R.) using random number generating statistical software, while another investigator assigned opaque envelopes (P.L.R.-S.) to each patient to be distributed within the sevoflurane or midazolam groups. . Furthermore, during data collection in the postoperative records, the results of these questionnaires were collected by two different researchers (C.M.M.-P., C.B.-D.).

How did you determine the sample size?

To calculate the sample size, the SPSS 30 program was used for an independent samples T-test, based on the hypothesis of seeing a difference of 0.14 between the groups with a deviation of 0.25, resulting in a sample of 52 patients per group (power of 80% and significance of 0.05).

Please report the articaine concentration.

Articaine 4%

Since you have followed the Consort Statement, you should provide the checklist as a supplementary file and add the Consort flow diagram in the text. The abstract should also be formatted according to these guidelines.

Please refer to de PDF document 2 attached.

What are the research limitations? Please report.

The following limitations have been included:

1.- The first of them would be to assess whether the difference in the route of administration of the sedative agent (intravenous vs inhalation) could change the results if the study were carried out between sedative agents administered in the same way. A typical example is some studies between propofol and midazolam that reach high levels of satisfaction.

2.- Another limitation that can be deduced is the type of surgical treatment performed in the present study. In Dentistry and in older adult patients, it is easy to find states of edentulism, which is why rehabilitation with osseointegrated implants is a frequent therapeutic option. In these cases, the manipulation of soft tissues and bone tissue is greater than that carried out in this research, so it would be desirable to be able to perform it in this indication, as authors such as Kawaai et al  and McCrea refer.

3.- A final limitation would be to consider whether these results could be similar, within this population group, if they were patients with cognitive disorders.

Conclusion: You should summarize the main findings in this section.

In summary, it can be generalized that sedation procedures can be very valuable tools in the field of oral surgery, since they offer safety for the patient, given the low incidence of adverse effects. Sedation with agents such as midazolam and sevoflurane appears to be quite safe given the absence of relevant hemodynamic changes. Midazolam has been shown to have a longer duration, as well as to develop greater digestive problems during the postoperative period. On the contrary, sevoflurane produced a greater sensation of dry mouth.  Both sedative agents have been perceived by the older adult population as satisfactory, recommending their use.

Reviewer 4 Report

Comments and Suggestions for Authors

Intravenous vs Inhalation Conscious Sedation in Oral Surgery: A Safety and Satisfaction Analysis in a Geriatric Population

Reviewer Report

Thanks to the authors for their study. The researchers aimed to compare conscious sedation with midazolam or sevoflurane used during oral surgeries in the elderly population. They reported finding significant differences between the two groups after oral surgery with intravenous and inhalation sedation, particularly noting that postoperative gastrointestinal disturbances and dry mouth were more prevalent in the midazolam group, while there were no differences between the two groups in terms of neurological symptoms. Furthermore, they emphasized that professionals administering conscious sedation methods should be well-trained in different sedative agents. Although the study presents important results in this regard, there are aspects that need improvement from both a scientific and formal perspective. My detailed comments:

Title: I suggest the title ‘Safety and Satisfaction Analysis of Intravenous and Inhalational Conscious Sedation in a Geriatric Population Undergoing Oral Surgery’ for a more fluent and academic heading.

Abstract:

Line 19: The patients’ mean ages and standard deviations, as well as their gender distributions, should be specified in parentheses.

Line 25-27: The slightly higher mentioned here should also include its statistical significance, or the p-value should be provided in parentheses.

Line 33-36: This conclusion could have been presented without the need for this study. Please provide more concrete clinical conclusions/interpretations/opinions that answer the research question based on the study results.

Introduction:

-        The introduction section should elaborate more on the necessity of oral surgery and why it is required in older age. The literature should focus more on the dental problems encountered in the later stages due to a lack of intervention in the early stages. Issues such as delayed dental fillings and root canal treatments leading to subsequent tooth loss, dental implants, and the challenges they present in advanced age, as well as the consequences of unaddressed orthodontic treatments leading to impacted or misaligned teeth causing bone loss and decay, should be discussed. Additionally, more background information should be provided on how periodontal pathologies are more challenging to treat in older age due to the lack of control over periodontal health. For further reference, you may consider reviewing the following studies:

https://doi.org/10.1111/jcpe.12681

https://doi.org/10.3390/children10060950

-        The rationale for conducting the study and its contribution should be emphasized further.

-        I also recommend adding a hypothesis and stating its acceptance or rejection based on the results in the discussion section.

Materials and Methods:

-        Why are lines 99-103 written in bold? This should be corrected.

-        How was the sample size calculation performed for the study? It should be stated by referencing a previous study.

-        The origins (references) of all the questionnaires used should be specified.

-        The tables should be prepared in accordance with the journal's template.

Results:

-        The abbreviations used in the tables should be specified below the table.

-        It has been reported that there is no difference in the characteristics mentioned in Table 5, but I see that a difference in weight has been found. Was this written incorrectly? It should be checked.

-        A p-value of (p=0.000) is not correct. Please use (p<0.001) instead.

Discussion:

-        Given the amount of data, this discussion is brief. Consider expanding it further. The suggested points for the introduction section can also be included here.

-        Highlight the strengths and limitations of the study.

-        Also, provide comments on the clinical integration of the results.

Conclusions:

-        Revision is also necessary here. Please stay focused on the main points of the study. Based on the results that response your research question, provide more concrete conclusions/interpretations/insights.

References:

-        The references are appropriate, but they should be expanded a bit more.

Comments on the Quality of English Language

Minor English language corrections required.

Author Response

Thanks to the authors for their study. The researchers aimed to compare conscious sedation with midazolam or sevoflurane used during oral surgeries in the elderly population. They reported finding significant differences between the two groups after oral surgery with intravenous and inhalation sedation, particularly noting that postoperative gastrointestinal disturbances and dry mouth were more prevalent in the midazolam group, while there were no differences between the two groups in terms of neurological symptoms. Furthermore, they emphasized that professionals administering conscious sedation methods should be well-trained in different sedative agents. Although the study presents important results in this regard, there are aspects that need improvement from both a scientific and formal perspective. My detailed comments:

Title: I suggest the title ‘Safety and Satisfaction Analysis of Intravenous and Inhalational Conscious Sedation in a Geriatric Population Undergoing Oral Surgery’ for a more fluent and academic heading.

The title has been change to the sugested:

Safety and satisfaction analysis of intravenous and inhalational conscious sedations in a geriatric population undergoing oral surgery

Abstract:

Line 19: The patients’ mean ages and standard deviations, as well as their gender distributions, should be specified in parentheses.

(Age: 84,00 ±8,410 / Gender: 17 men/36 women) and inhalation conscious sedation (Age: 82,73± 8,809 / Gender: 18 men/ 33 women).

Line 25-27: The slightly higher mentioned here should also include its statistical significance, or the p-value should be provided in parentheses.

Anxiety levels were slightly higher in the midazolam group (MDASscore: 9.60 ± 5.849; HARSscore: 27.19 ± 6.802) compared to the sevoflurane group (MDAS score: 8.37 ± 5.063; HARSscore: 24.92 ± 5.199), with no statistical significance (MDAS, p= 0.453/ HARS, p= 0.065).

Line 33-36: This conclusion could have been presented without the need for this study. Please provide more concrete clinical conclusions/interpretations/opinions that answer the research question based on the study results.

Conclusions: Sedation with agents such as midazolam and evoflurane seems to be quite safe given the absence of relevant hemodynamic changes. Midazolam has been shown to have a longer duration, as well as to develop greater digestive problems during the postoperative period. On the contrary, sevoflurane produced a greater sensation of dry mouth.  Both sedative agents have been perceived by the older adult population as satisfactory, recommending their use.

Introduction:

-        The introduction section should elaborate more on the necessity of oral surgery and why it is required in older age. The literature should focus more on the dental problems encountered in the later stages due to a lack of intervention in the early stages. Issues such as delayed dental fillings and root canal treatments leading to subsequent tooth loss, dental implants, and the challenges they present in advanced age, as well as the consequences of unaddressed orthodontic treatments leading to impacted or misaligned teeth causing bone loss and decay, should be discussed. Additionally, more background information should be provided on how periodontal pathologies are more challenging to treat in older age due to the lack of control over periodontal health. For further reference, you may consider reviewing the following studies:

https://doi.org/10.1111/jcpe.12681

https://doi.org/10.3390/children10060950

Population aging is an unquestionable reality, which, as Hazara [1] points out, is expected to reach 22% by the year 20250. This increase will be accompanied by a greater prevalence of systemic diseases and pharmacological treatments that make the older adult population more vulnerable to oral diseases such as periodontal disease, dental caries and even precancerous lesions [2,3].

Coll et al [4]., report that data collected in 2020 by the National Center for Health Statistics of the United States showed that the prevalence of root cavities was more than double in older adults (36%) compared to older adults. Young people Likewise, the prevalence of periodontitis also increased with age, reaching 64% of older adults with moderate or severe periodontitis, compared to less than 38% in younger patients.

Although aging has no or very limited effects on the treatment of dental cavities and periodontal diseases, it is important to note that there are different factors, such as the use of medications, reduced salivary secretion, deteriorated fixed and removable dental prostheses, and the decrease in the ability to perform effective oral hygiene, which can have direct consequences on the results and consequently lead to tooth loss, becoming one of the common treatments in this population group [5,6].   

Tooth extractions may differ in terms of technique and the appearance of complications, and different population groups must be taken into consideration. Older adults, due to their comorbidities, would present greater risks in cases of pathologies such as diabetes or in situations of impacted teeth [7,8].

-        The rationale for conducting the study and its contribution should be emphasized further.

In the dental field, interest in conscious sedation is increasing across different population groups, particularly in the geriatric population due to the aging of the general population. In this regard, scientific literature specifically comparing the use of inhalation sedation with sevoflurane and intravenous sedation with midazolam in geriatric populations is limited. Thus, the conduct of this study is justified.

-        I also recommend adding a hypothesis and stating its acceptance or rejection based on the results in the discussion section.

It was proposed as a null hypothesis that sedation between both methods does not present differences in terms of complications and the degree of satisfaction perceived by patients.

Materials and Methods:

-        Why are lines 99-103 written in bold? This should be corrected.

                  This has been corrected.

-        How was the sample size calculation performed for the study? It should be stated by referencing a previous study.

To calculate the sample size, the SPSS 30 program was used for an independent samples T-test, based on the hypothesis of seeing a difference of 0.14 between the groups with a deviation of 0.25, resulting in a sample of 52 patients per group (power of 80% and significance of 0.05).

-        The origins (references) of all the questionnaires used should be specified.

They have been specified.

-        The tables should be prepared in accordance with the journal's template.

                  The document will go through editing services in to meet all the requirements.

Results:

-        The abbreviations used in the tables should be specified below the table.

                  This has been added.

-        It has been reported that there is no difference in the characteristics mentioned in Table 5, but I see that a difference in weight has been found. Was this written incorrectly? It should be checked.

                  Checked.

-        A p-value of (p=0.000) is not correct. Please use (p<0.001) instead.

                  Changed

Discussion:

-        Given the amount of data, this discussion is brief. Consider expanding it further. The suggested points for the introduction section can also be included here.

-        Highlight the strengths and limitations of the study.

Some limitations must be established to evaluate whether these results can be maintained. The first would be to assess whether the difference in the route of administration of the sedative agent (intravenous vs inhalation) could change the results if the study were carried out between sedative agents administered in the same way. A characteristic example are some studies between propofol and midazolam that reach high levels of satisfaction [57-59].

Another limitation that can be deduced is the type of surgical treatment performed in the present study. In Dentistry and in older adult patients, it is easy to find states of edentulism, which is why rehabilitation with osseointegrated implants is a frequent therapeutic option. In these cases, the manipulation of soft tissues and bone tissue is greater than that carried out in this research, so it would be desirable to be able to perform it in this indication, as authors such as Kawaa ie tal and McCrea refer.

-        Also, provide comments on the clinical integration of the results.

In this way and as a summary, it can be generalized that sedation procedures can be very valuable tools in the field of oral surgery, since they offer safety for the patient, given the low incidence of adverse effects. The older adult population may be particularly complex in many cases due to systemic and cognitive conditions, which could benefit from sedation treatments. It is essential that professionals applying conscious sedation methods are well versed in the guidelines for the use of different medications and are properly trained to administer them safely and address potential adverse events.

Conclusions:

-        Revision is also necessary here. Please stay focused on the main points of the study. Based on the results that response your research question, provide more concrete conclusions/interpretations/insights.

5. Conclusions

Sedation with agents such as midazolam and sevoflurane appears to be quite safe given the absence of relevant hemodynamic changes. Midazolam has been shown to have a longer duration, as well as to develop greater digestive problems during the postoperative period. On the contrary, sevoflurane produced a greater sensation of dry mouth.  Both sedative agents have been perceived by the older adult population as satisfactory, recommending their use.

References:

-        The references are appropriate, but they should be expanded a bit more.

                  References were added as suggested also by other revisors.

Round 2

Reviewer 3 Report

Comments and Suggestions for Authors

The manuscript has been substantially improved. However, there are still some unclarities. The article type is not included in the title. In addition, the authors have provided the IRB number, but the registration number is still lacking. Whether the study has been registered in a recognized public registry such as PROSPERO remains unclear.

Author Response

We have added the article type on the first page. We apologize for not doing so earlier.

With regard to the authorization of the Ethics Committee, the data we have offered are the only ones provided by the Committee of the University Hospital of La Paz itself, which we included. The registration in PROSPERO has not been assessed as we understand that it is a registry for systematic reviews.

Reviewer 4 Report

Comments and Suggestions for Authors

The revisions have been significantly completed; however, I still have a few minor suggestions:

-        Introduction section has been expanded but is presented in a manner resembling the Discussion section. The narrative flow should be adjusted to align with the purpose of the Introduction. Additionally, the orthodontic problems mentioned in the previous report should also be addressed.

-        In the Discussion section, there are two separate paragraphs starting with 'In summary' and 'To summarize.' These should be merged or modified to ensure a coherent flow.

-        The acceptance or rejection of the null hypothesis should be stated in the Discussion and/or Conclusion section based on the results.

Comments on the Quality of English Language

Minor English language corrections are required.

Author Response

1.- Introduction:

Regarding the article that you suggested to appear in the introduction (https://doi.org/ 10.3390/ children10060950); unless mistaken, it corresponds to:

Investigation of the Relationship of Impacted Maxillary Canines with Orthodontic Malocclusion: A Retrospective Study. Orhan Cicek, Turhan Gurel and Busra Demir Cicek.

 All the authors of this publication have analyzed this publication separately and we do not see the relationship it may have with our research, so we would ask you, if possible, to provide us with more details.

2.- Discussion

Fixed the repetition of  ”In summary” and “To summarize”

3.- Null hypothesis

The confirmation of the null hypothesis has been added in the conclusions section.

4.-Quality of English Language:

We attach the document that certifies that the English translation of this article has been carried out by the services of MDPI.
